# Estimating turnover and industry longevity of Canadian sex workers

**Lynn Kennedy** *

Sex Work Population Project, Vancouver, British Columbia, Canada

* lynn.kennedy@populationproject.ca

**Editor:** Hamid Sharifi, HIV/STI Surveillance Research Center and WHO Collaborating Center for HIV Surveillance, Institute for Future Studies in Health, Kerman University of Medical Sciences, ISLAMIC REPUBLIC OF IRAN

**Data Availability Statement:** The data is held in a public repository which can be found via the url https://osf.io/jx7ch/.

## Abstract

How long indoor sex workers stay employed in collectives is a poorly understood aspect of sex worker agency in industrialized democracies. This study provides estimates of turnover, the rate at which workers leave employment, using a subsample of 76 collectives representing 3545 workers over a one-year period. All the collectives provided data on individual workers via external websites. The collectives were identified in a larger random sample of 783 advertisers from a popular Canadian classifieds site used by sex workers, all of whom provided URLs as part of their ad contact information. Monthly between October 2022 and October 2023, individual workers associated with the subsample of advertisers were identified from web pages maintained by these advertisers and scheduling data was collected where available. Worker turnover was estimated based on whether workers were visible one month to the next. Over the year, estimated turnover ranged from 12.0% to 16.0% (mean 14.2% SD 1.1%). Turnover was not affected by month or number of workers in the collectives. Mean 41.1% workers (SD 23.5%, N = 51 advertisers) were scheduled on any given day. Workers were visible for a mean 5.5 months (SD 4.5) with those visible for one month being the largest single group. Most sex workers in collectives are likely not permanent full time employees, and the extremely brief work histories of many suggest that failure in the industry may be common for this subpopulation.

## Introduction

As the early Canadian sex worker rights activist Alexandra Highcrest described it in 1990, "The sex trade industry is, for many, a transient occupation." [1,2] In industrialized democracies, sex workers who meet clients in person exercise agency in spite of significant structural barriers [3–7]. Some research suggests that many workers go "offline" regularly [8,9]. There is clearly a need to quantify how common this is. An indirect indication of this labor market volatility is the difficulty that researchers have in contacting sex workers. For example, in Nelson et al. [9], where 1730 independent escort websites were originally found only 839 could be contacted after a 4-month period. Similarly, a survey by Cunningham and Kendall of sex workers who had profiles on a large US review site [10], out of 26189 emails sent, only 13333 were deliverable.

**Funding:** The author(s) received no specific funding for this work.

**Competing interests:** The authors have declared that no competing interests exist.

One possible reason for attrition in the industry is that many sex workers have other, more permanent employment. Many sex workers find combining sex work with non-sex work ("straight work") to be attractive, given the flexibility and relatively high pay of sex work [11]. In the Cunningham and Kendall survey [10] 43.2% of the 685 respondents (linearized SE 0.037) had straight employment. These respondents worked at straight employment on average 28.2 hours per week (linearized SE 1.2) [12]. Notably, the number of hours spent in sex work for the respondents was likely far less, as they reported seeing 5.5 clients per week on average (linearized SE 1.1) with an average session length of 118 minutes (linearized SE 11). Similarly, Bowen [13] found that among a group of 22 current and former sex workers, 5 were concurrently engaged in both straight work and sex work, 8 were out of sex work but would consider re-entering, and 9 were no longer active and did not plan to return. These statistics challenge the notion of sex workers having a simple binary relationship with the industry.

Pitcher [14] interviewed a sample of 36 indoor sex workers in depth and observed three broad patterns of involvement in sex work: interim pathways, multiple transitions, and longer-term careers. Interim pathways participants were divided between those who did sex work according to a plan for a specific period of time, versus those who did it while considering other options. Multiple transitions participants were those moving in and out of sex work and consisted of those who were divided between the undecided, those who planned a return, and those who had parallel careers. Participants in the Bowen and Pitcher studies [13,14], with one exception, had been in the industry for more than one year.

The relatively long work histories of the participants seen in [13,14] begs the question: how many workers are in the industry for less than one year? This study builds on previous work by the author [15] that considered how sex worker populations changed over time by analyzing advertising data from six prominent Canadian classified advertising sites commonly used by sex workers. Typically, advertisers on these sites did not advertise for long periods of time (mean of 73.3 days, SD 151.8). In that study, as in this, advertisers were the identified authors of groups of ads which might represent one or more workers. One difficulty with this study was that it only considered advertisers and workers represented in online classified ads. Workers associated with collectives might not have been adequately described in these ads, for example, and may have different work histories than independent advertisers.

It is risky to collectivize in a criminalized environment where earning money from and advertising sexual services are both illegal. This was the case in Canada in 2022 [16], where the goal of legislation was the eradication of the sex industry [17]. Nevertheless, many workers continue to work collectively: this was the case for most of the workers associated with the sampled advertisers in this study, with the majority engaging with third parties. Bruckert and Law [18] reported that the main reasons for working collectively were the comfort of being able to work from home, avoiding police harassment, sense of community, enhanced security protocols, sober working environment, improved financial security, having someone else obtain clients, and allowing a better work-life balance. Many of the workers described in Bruckert and Law worked both independently and with third parties, with some moving between the two as needed.

The relationship between a worker and a third party can take a number of forms. Among the advertisers described below, there were examples of more formal business establishments such as spas and escort agencies. There was also a more informal ecosystem of worker collectives and third parties who provide services such as fielding calls, providing collective advertising, or safe workspaces described in Bruckert and Law as *associate relationships* [18].

The advertisers in this study all provided some form of external website as a contact, and many of these were independently owned. Collectives that maintain their own websites often provide detailed information on workers, including daily schedules. Furthermore, most of

these websites were actively updated during the study period: typically on a weekly basis. This external website data made it possible to not only measure worker availability from month to month, but also estimate how many workers were active on any given day. This data makes it possible to develop a more precise understanding of the work histories of workers associated with these advertisers.

## Study objectives

The main objective of this study is to better understand employment volatility among indoor sex workers. Employment volatility was measured in three ways. Firstly, the turnover rate for workers working in collectives was estimated month to month. Secondly, how long workers remain associated with any collective was estimated. Thirdly, collective advertisers were identified who were actively seeking new workers. Lastly, how long advertisers had been in business was estimated using external website data.

The second objective sought to determine characteristics of the sampled advertisers. For all sampled advertisers, the number of workers, including the number who were working collectively, was estimated from both ad data and external website data. In addition, for collectives, the geographic locations, worker genders, clientele and business types were identified.

## Materials and methods

This study uses a combination of qualitative and quantitative techniques to provide evidence for employment volatility based on a group of sex work advertisers who advertised on external websites in addition to using classified advertising. Advertisers were identified from ad data collected from classified ads. Ads were downloaded between September 15, 2021 and September 22, 2022 from one prominent Canadian classified advertising site. The site in question has been described before [15] and is one of the sites identified by industry experts from the *Sex, Power, Agency, Consent, Environment and Safety Project* [19] as an advertising venue commonly used by sex workers. SPACES was initiated in 2012 at the University of British Columbia to explore health and safety issues experienced by these workers.

Ads were identified in site maps provided by the classifieds site. The sitemaps were XML format files that were updated daily, containing ad URLs from the site. Sitemaps are provided by website operators to facilitate search engine indexing. Sitemaps from the classifieds site were downloaded three times per day in 2021–2022, and ad URLs were extracted using a custom PERL script [see "downloaders" in 20]. The discovered ad URLs and any associated images were then downloaded and stored for later analysis.

On the classifieds site, advertisers could be identified by chat names used for an internal chat function. In addition to ads, many advertisers maintained profile pages on the classifieds site which were easily discovered, when available, from the chat name. Advertisers could optionally provide contact data in their ads or profiles, which often contained links to external websites. However, not all advertisers identified from the downloaded ads represented sex workers who meet clients in person. To eliminate irrelevant advertisers, the three most commonly posted ads for each advertiser were visually inspected and advertisers were included if they represented a sex worker that provided services in person with contact information. These advertisers represented the initial sample population for the study.

For each ad, the first time the ad had been seen and the last time the ad had been seen was recorded. Ads were associated with advertisers and this data was stored in a MariaDB database [21].

Advertisers qualified for the study if they had been actively advertising between August 23, 2022 and September 22, 2022, and had used a URL as part of their contact details. An initial

random selection of 1000 advertisers was selected from this sampling frame. Advertisers that had either live ads or profile pages on the classified ad site between September 23 and September 30, 2022 were included as the initial sample. Web pages from contact URLs provided by these advertisers were downloaded between October 1 and October 8, 2022. The Firefox web browser [22] was used to save web pages and associated files and, where feasible, the web page layouts were captured as images. This initial data set is described in more detail in [23 preprint].

All the collective advertisers from the original sample who listed individual workers on their websites were identified. Web pages listing workers and, when available, pages with worker schedules were downloaded monthly between October 1, 2022 and October 8, 2023 and unique worker names were collected. Data such as the date, website, worker names, number of workers, and number of scheduled workers was stored in a MariaDB database [21]. An anonymized copy of this database is available in supplemental materials S1 File. Fig 1 shows the name extraction process from start to finish.

Name lists for each advertiser were compared month to month. The monthly variable *missing* or names visible in the previous month not visible the following month, an estimate of workers who left that month, was calculated. For any given month, advertisers who did not have valid web domains for that month or the following month were excluded from the turnover calculations.

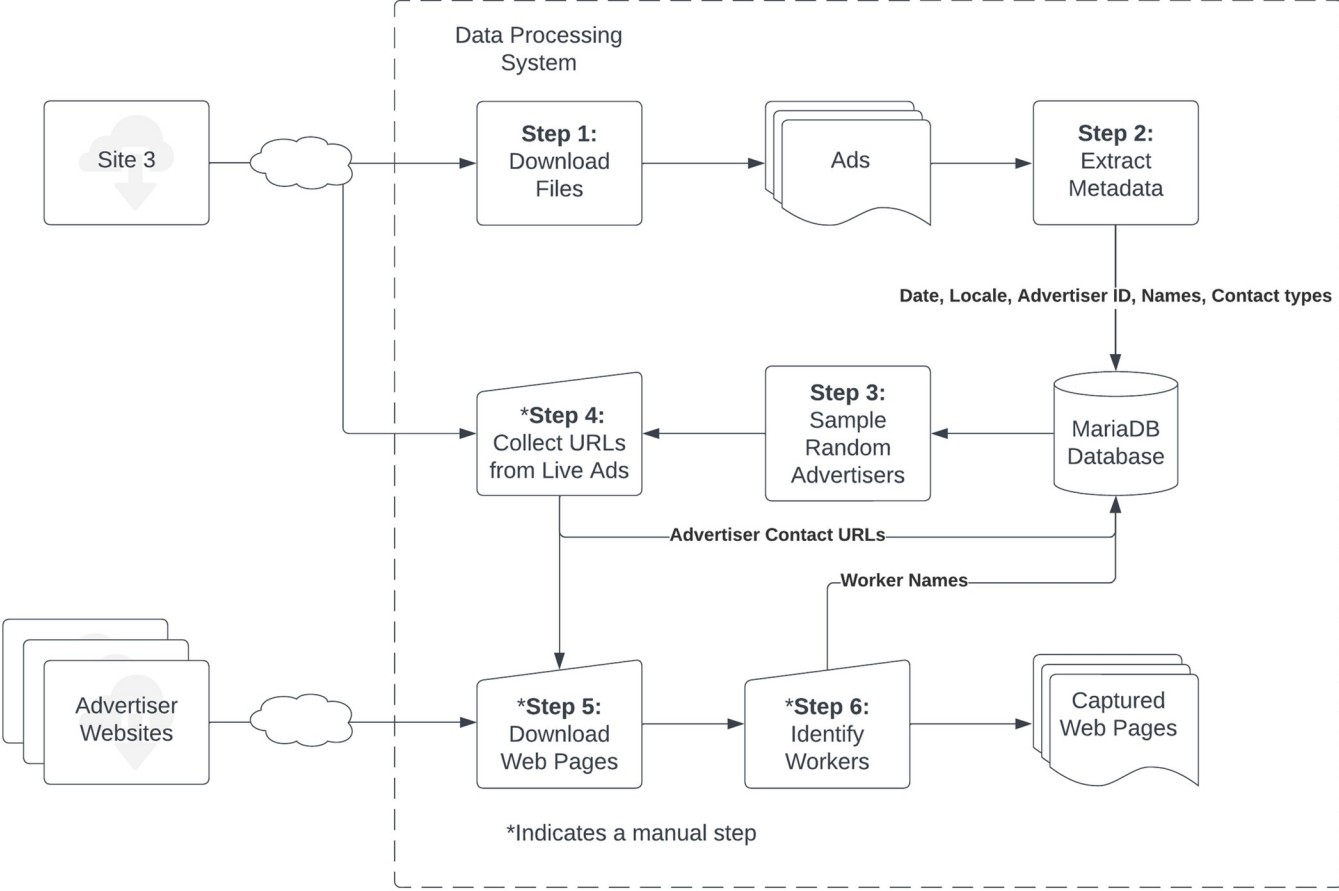

**Fig 1. Name extraction process.** Data was collected and processed in six stages. Steps 5 and 6, *Download Web Pages* and *Identify Workers*, were repeated monthly for the subsample of collective advertisers who provided information on individual workers on their websites.

The subsample was analyzed for other variables. Firstly, the advertisers were stratified by province based on addresses and phone numbers listed on their external web pages. Secondly, advertisers were identified who discussed employment or had hiring web forms on their websites. Thirdly, advertisers were identified who employed cis-male or trans-female workers.

Fourthly, advertisers were identified who accepted women or couples as clients. Finally, to see if the original group of downloaded advertisers were still actively advertising, sampled advertisers were searched for in a set of ads downloaded during the last two months of data collection.

Advertiser years in the industry were estimated from external website data from the full random sample of advertisers. This evidence consisted of either mentions of how long the advertiser had been in business or a year, for example a copyright year, found on a web page for individual advertisers. The oldest year found for each advertiser who provided this information was recorded.

## Statistical measures

Descriptive statistics were calculated for the subsample of collectives tracked from October 2022 to October 2023. These included the average number of workers per collective, the number of collectives in each province, the number of collectives who were actively hiring, and the total number of months individual workers were visible. For each of the tracked collectives, monthly turnover was calculated based on the following standard formula [24–27]:

$$turnover = \frac{missing}{\left[\frac{(P_{initial} + P_{final})}{2}\right]} \tag{1}$$

*Missing* is the number of workers in the initial population at the start of the month that were not found in the final population at the end of the month. *Turnover* is the proportion of *missing* divided by the average of the initial and final population for any given month. A code listing for the turnover calculations can be found in supplemental S2 File. In addition, the total number of workers seen monthly and the number of workers active on the nearest day to the day of data collection were also calculated. The Pearson correlation between *turnover* and number of monthly workers was calculated to see if competition affected the number of workers leaving; the Pearson correlation between *turnover* and month was calculated to see if there were seasonal relationships that affected the number of workers leaving. Correlations were calculated using the R *cor.test* function [28] using a 95% confidence interval on bimonthly pairs of advertiser worker counts, where workers could be counted for both months.

For all advertisers, in the year prior to the September 2022 sampling period, the number of ads produced by each advertiser was counted. The length of time each advertiser had been advertising was calculated as the number of days between when the first and last ads were seen. These statistics were stratified based on inclusion/exclusion in the initial sample and the subsample. In addition, for the initial sample, statistics on estimated years active in the industry were calculated for advertisers providing this information. Years active statistics were stratified by social context, or whether an advertiser represented a collective or an individual.

## Ethics statement

All source data used in this study consisted of publicly available data at the time it was collected and was collected in accordance with the policies of the sites in effect at the time. The methods used are conformant with the ethical standards of the Canadian Sociology Association (section 4.10 II) and the American Sociology Association (section 10.5 c) [29,30]. As the replicability of

the main results of this paper is important, a data set is provided as part of the supporting information. However, to protect the safety and privacy of advertisers and third parties, all identifying information has been removed, including the names of the source websites.

## Results

### Sample selection

Between September 15, 2021 and September 22, 2022, 1217296 classified ads were downloaded from one prominent Canadian classifieds site used by sex workers. The downloaded web pages contained 54558 chat names that identified advertisers who had authored one or more ads. From this original group, 39562 advertisers were included. The 14996 excluded advertisers comprised 11962 who were clients seeking services, 1718 with no contact information, 845 who represented "hookup" sites, 457 representing other services (drivers and other related services), and 14 who offered non-sexual therapeutic services. A large sample of classified ads (N = 891695) associated with the included advertisers was used to identify advertisers who included URLs as part of their contact information.

Advertisers active between August 23, 2022 and September 22, 2022 who had included a URL in their contact information either to an external website or a profile page were the sampling frame for inclusion in the study. There were 2452 qualifying advertisers from which 1000 were randomly selected. Of the selected advertisers, 783 had readable contact information from recent ads or profile pages at the time of initial data collection. These advertisers comprised the initial sample used for the study.

### Working collectively

Advertisers could represent one or more workers. Workers were identified using names from both classified ads and external website data. After correcting for advertisers who were using the same external web domain (N = 35 representing 12 domains), 3189 workers were estimated to be represented in the sample of 783 advertisers using unique names found in both ads and external websites. Out of the 3189 identified workers, 1448 workers (45%), mostly associated with collectives, appeared to be unique to the external websites. There were 102 advertisers, representing 1941 workers, who explicitly advertised as some form of collective via external websites. These organizations consisted of 57 spas, 28 escort agencies, 12 associate relationships, and 5 worker collectives based on website descriptions.

A subsample of 76 collective advertisers which comprised all the advertisers who provided detailed information on individual workers was tracked monthly. Workers were overwhelmingly identified as cis-female in the tracked subsample; however, eight advertisers had at least one male worker and four had at least one trans female worker during the data collection period. Only one advertiser appeared to be exclusively representing trans women. Nine advertisers were associated with workers who accepted women or couples as clients.

In the year prior to the sampling period, advertisers who were part of the initial random sample advertised longer and produced more ads (N = 783; mean 335 days, SD 98; mean 93 ads, SD 431) than the other classified advertisers who did not use URLs in their contacts (N = 34683; mean 205 days, SD 124; mean 18 ads, SD 86). The subsample was similar to the initial random sample but produced more ads, advertising for a mean 348 days (SD 94, N = 76) and producing a mean 162 ads (SD 265) during this period.

In any given month between October 1, 2022 and October 8, 2023 the subsample of 76 collectives were associated with median 17.5 workers (IQR 9.1–22.7, mean 21.4, SD 20.7). Fig 2 is a histogram of the average number of workers per month per advertiser. The majority of advertisers were located in Ontario (N = 27) and British Columbia (N = 21) followed by

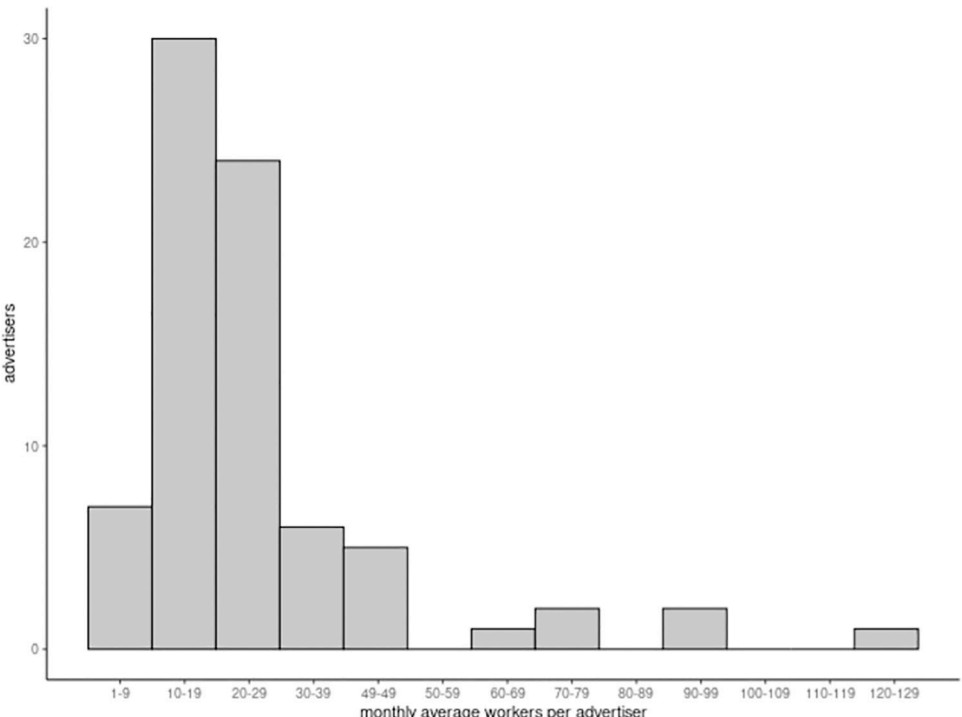

**Fig 2. Average monthly workers per collective between October 1, 2022 and October 8, 2023.**

Quebec (N = 13), Alberta (N = 11), Saskatchewan (N = 2) and Newfoundland (N = 1). Two advertisers did not have a fixed business location based on website information. Most advertisers (88%, N = 67) used their websites to recruit new workers during this period.

## Worker turnover in collectives

Table 1 compares the monthly turnover rates of the tracked advertisers with the job vacancy rate for all Canadian workers [31] between October 1, 2022 and October 8, 2023. Over the

**Table 1. Worker turnover from October 1, 2022 to October 8, 2023.** "Missing" refers to workers seen in a given month not seen in the subsequent month. Turnover is calculated using Formula (1). The Canadian job vacancy rate is provided for reference from Statistics Canada data.

| month | advertisers | workers | missing | turnover | Canadian job vacancy rate |
|---|---|---|---|---|---|
| 2022–10 | 76 | 1441 | 215 | 14.21% | 5.5% |
| 2022–11 | 75 | 1584 | 212 | 13.10% | 5.0% |
| 2022–12 | 75 | 1653 | 232 | 14.14% | 4.4% |
| 2023–01 | 75 | 1628 | 198 | 12.00% | 4.6% |
| 2023–02 | 72 | 1671 | 234 | 14.18% | 4.5% |
| 2023–03 | 72 | 1629 | 253 | 15.65% | 4.7% |
| 2023–04 | 73 | 1605 | 243 | 15.07% | 5.0% |
| 2023–05 | 72 | 1619 | 213 | 13.06% | 4.8% |
| 2023–06 | 68 | 1642 | 250 | 15.17% | 4.5% |
| 2023–07 | 69 | 1653 | 221 | 13.51% | 4.1% |
| 2023–08 | 66 | 1618 | 223 | 13.85% | 4.1% |
| 2023–09 | 64 | 1603 | 253 | 15.97% | 3.9% |
| 2023–10 | 64 | 1566 | | | |

year, estimated turnover rates ranged from 12.0% to 16.0% (mean 14.2% SD 1.1%). On the nearest day to the sample date, advertisers who posted worker schedules had scheduled mean 41.1% workers (SD 23.5%, median 34.8%, IQR 25.0%-50.0%, N = 50). The Pearson correlation between turnover and month was not significant (p = 0.38) nor was the correlation between turnover and monthly workers (p = 0.97) for 869 pairs of monthly advertiser worker counts.

In September and October 2023 62 of the original 76 advertisers were still advertising on the classifieds site. Of these, there were 12 advertisers who were still actively advertising after their associated domain had become inaccessible. Conversely, there were 8 advertisers with active domains who were not found in the classified ad data at that time.

Five advertisers did not change worker listings over the study period. A possible reason for this is that they may not have updated their websites over the year they were tracked. However, dropping these advertisers had no significant effect on the turnover calculations.

## Worker longevity in collectives

A total of 3545 workers were identified between October 1, 2022 and October 8, 2023 in the sub-sample of 76 collectives. The number of months a worker was associated with a collective was estimated by counting the number of months each unique name was visible on external websites maintained by the advertisers during the study period. Only advertisers who had accessible external web domains for the full study period were included: 62 advertisers representing 3226 workers. Fig 3 is a histogram showing how long workers were associated with each advertiser. Most workers (N = 827) were only visible for one month. However, there may have been workers who were active before the first month or after the last month. On average, there were 51 of single-month workers (N = 51) per month for the eleven months between the first and last month of data collection. Multiplying this by the number of months in the data collection period (N = 13), suggests that the actual number of workers only appearing for one month would be closer to 658 (20%). Each name was visible for mean 5.5 months (SD 4.5, median 4, IQR 1–9).

## Advertiser years in the industry

The high attrition rate for many workers in collectives is in contrast to the longevity of many of the advertisers using contact URLs. Most of the original sample (78% = 611/783) were still actively advertising six months after September 2022. Additional evidence for time in the industry was found for 231 advertisers based on captured website data. This data provides a lower bound on how long these advertisers claim to be in the industry. Fig 4 shows this distribution. Collective advertisers (N = 39) were found to be active at least median 4 years (IQR 3–10, mean 7.7, SD 7.2). This was considerably longer than the independent advertisers who were active at least median 2 years (IQR 1–4, mean 3.9, SD 4.6).

## Discussion and conclusions

Sex workers associated with a sample of 76 online advertisers were tracked over a one-year period. This group of advertisers provided detailed information on workers associated with them, often including daily schedules. These collectives tended to be relatively small, with a median 17.5 workers each month per collective. The collectives consisted almost entirely of cis-female workers. Some advertised services for women and couples (12% = 9/76).

Short term employment appeared to be the norm rather than the exception for workers who were associated with these collectives. Mean turnover for the collectives ranged from 12.0% to 16.0% monthly (mean 14.2%, SD 1.1%). Turnover was not affected by the number of workers in the collectives, nor was it affected by seasonal variation. The estimated turnover rate for sex workers in collectives was considerably higher than the monthly vacancy rate for

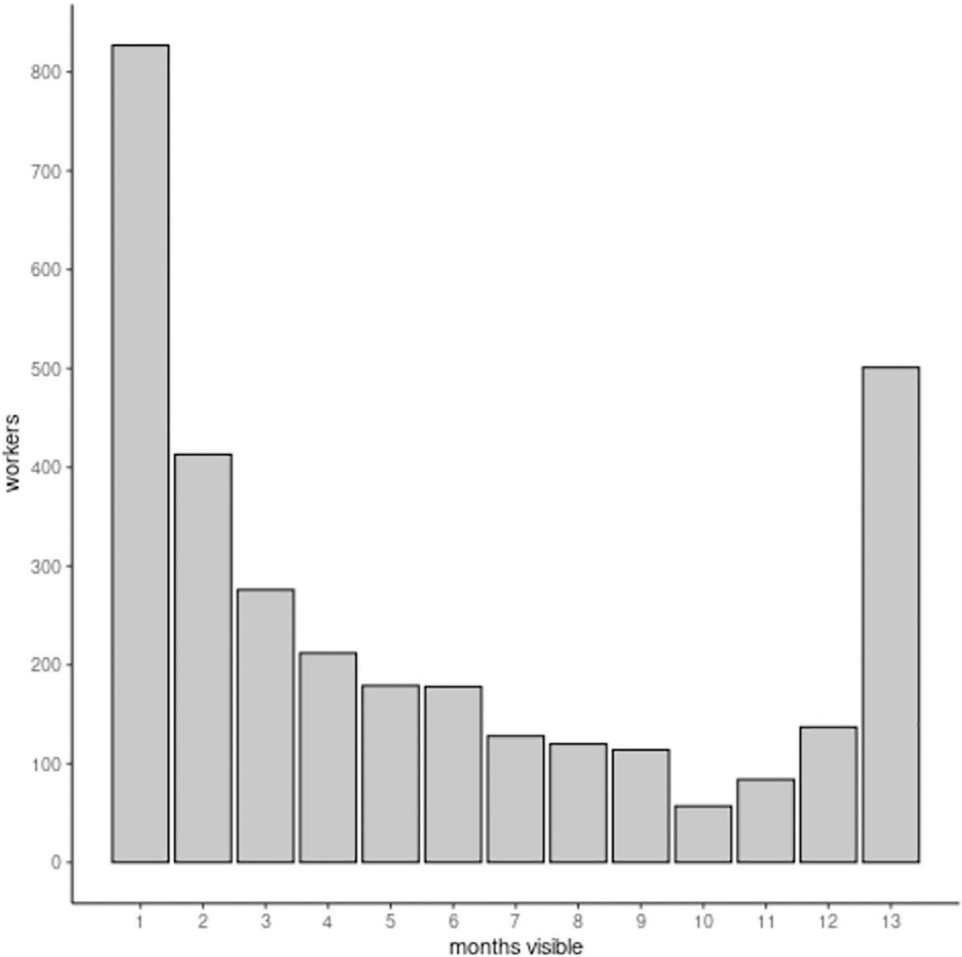

**Fig 3. Length of time that workers were visible in website listings for 62 collectives between October 1, 2022 and October 8, 2023.**

Canadian workers during the same period (mean 4.7%, SD 0.4%) [31]. Furthermore, 88% of advertisers were actively looking for workers throughout the year, suggesting that high employee turnover was an ongoing problem for many of these advertisers. Workers were estimated to be employed for a mean 5.5 months (SD 4.5), with the mode being 1 month. The distinct "U" shape of this distribution was striking, showing a distinct cutoff between long-term and short-term workers at around 10 months.

This subsample was part of a larger random sample of 783 advertisers selected from advertisers on a popular Canadian classifieds site used by sex workers. These advertisers all provided links to external websites or profile pages in their contact information. These links were used to gather additional information about each advertiser. This group tends to advertise longer and produce more ads than other advertisers on this classifieds site, and many in the larger sample indicated being in the industry for multiple years. However, using sitemaps to discover ads may have included ads that, while live at the time of collection, were not easily visible to site visitors. In view of this fact, it is possible that advertisers may have been active for less than the time indicated by discovered ads. See supplemental materials S1 Appendix for a discussion.

Around two thirds of the 3189 workers identified in the larger sample were associated with a relatively small group of 102 collective advertisers. The subsample of 76 advertisers used for

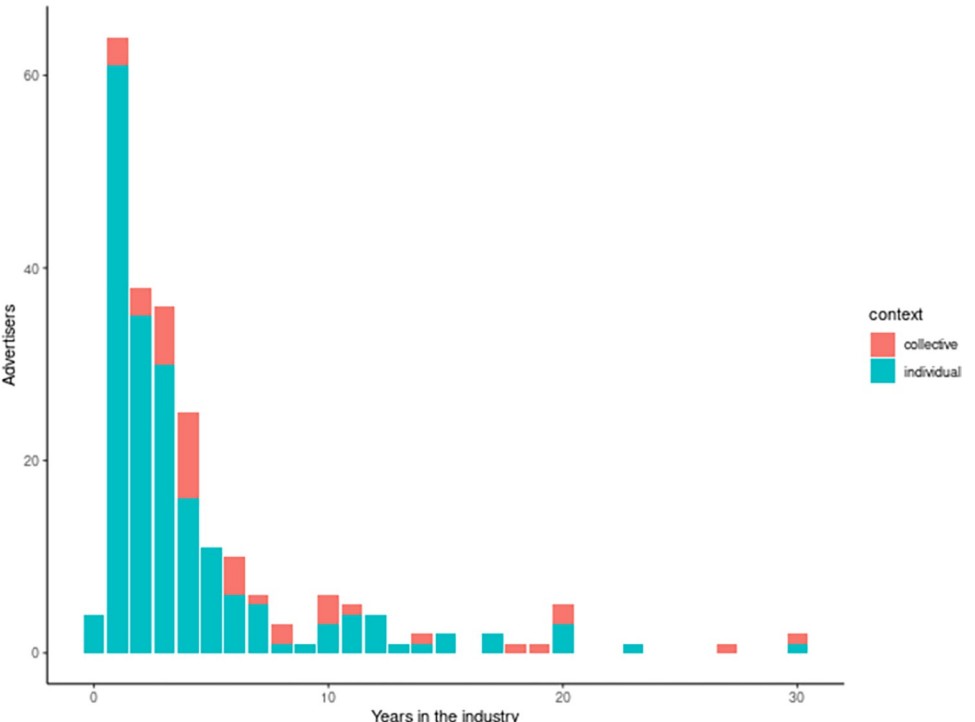

**Fig 4. Estimated advertiser years in the industry from external website data by social context.** The years represent the minimum years in the industry for the advertiser based on website data from 231 advertisers in October 2022.

turnover calculations were associated with between 1441 and 1671 workers over the year they were tracked (mean 1612, SD 59). The external website listings of workers were found to be far more reliable than the advertising data. Of the workers identified in October 2022, 45% were only visible on the external websites.

It is possible that workers who are part of collectives are in collectives because they either do not plan to be in the industry for a long time or switch between working collectively and advertising independently [13,14,18]. However, the extremely brief employment duration of many workers seen in this study suggests that some may in fact not be successful, a phenomenon seen elsewhere in the literature on street involved sex workers [32]. It is possible that the reason this has not been widely reported before may be the result of the difficulty in recruiting research participants who have industry involvement of such short duration. Advertisers who did not use contact URLs appear to have advertising histories similar to the workers in the tracked collectives, suggesting that this pattern may be common. Further research is needed to determine the rate at which workers fail, quit, switch work venues, or go independent.

The tracked collectives may over-represent some regions of Canada. However, given that the subsample of collective advertisers were selected from a large random sample, this over-representation may reflect regional differences, such as licensing requirements, demand for services, or other factors that may affect how sex workers organize.

## Conclusions

Researchers using advertising data alone to characterize sex worker populations may not identify all active workers for a given time period, given the fact that many of the identified workers in this study were not visible in advertising data. Furthermore, the reality that many workers

do not remain at any given work venue for long periods of time presents challenges for researchers who want to find representative samples of this subpopulation. While using advertising data as a sampling frame is likely to produce samples that are more representative of the general sex worker population, it is not enough. Direct outreach to collectives is likely also important. To achieve sufficient detail, it may be necessary for researchers studying collectives to employ embedded ethnographic techniques similar to those used for studying street involved workers [32,33].

The collective advertisers identified in this study represented a variety of different organizational structures, and some appeared to combine elements of independent and collective organization (associate relationships). Which arrangements provide the most benefit for workers is an open question. Most likely this will depend, among other considerations, on how long workers plan to be in the industry.

The data collected here shows that the reality of sex work in Canada is that the majority of workers are not permanent full time employees. Some workers may in fact not succeed in this very competitive industry, which also appeared to be the case for some advertisers, a phenomenon reported previously [8]. Profit is not guaranteed. More specifically, the evidence presented shows that there are at least two distinct populations among indoor workers based on how consistently they are employed. This echoes trends in the Canadian labor market, where the number of workers in the casual "gig" economy has increased in recent years [34] and, arguably, most of the workers identified in this study would best be described as participating in this economy. Mentoring and support for these workers would likely be a better fit than exiting programs, as the majority of these workers may be out of the industry in less than a year. Decriminalization is clearly only a first step in rethinking this labor market in a way that protects the rights of these workers.

## Supporting information

**S1 File. MariaDB database containing monthly name data.**
(ZIP)

**S2 File. Code listing for turnover calculation.**
(ZIP)

**S1 Appendix. Comparison of data collection techniques.**
(DOCX)

## Author Contributions

**Conceptualization:** Lynn Kennedy.

**Data curation:** Lynn Kennedy.

**Formal analysis:** Lynn Kennedy.

**Funding acquisition:** Lynn Kennedy.

**Investigation:** Lynn Kennedy.

**Methodology:** Lynn Kennedy.

**Project administration:** Lynn Kennedy.

**Resources:** Lynn Kennedy.

**Software:** Lynn Kennedy.

**Supervision:** Lynn Kennedy.

**Validation:** Lynn Kennedy.

**Visualization:** Lynn Kennedy.

**Writing – original draft:** Lynn Kennedy.

**Writing – review & editing:** Lynn Kennedy.

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
