## [Decision Letter · Decision Letter 0]

28 Dec 2023

PONE-D-23-35872Estimating turnover and industry longevity of Canadian sex workersPLOS ONE

Dear Dr. Kennedy,

Thank you for submitting your manuscript to PLOS ONE. After careful consideration, we feel that it has merit but does not fully meet PLOS ONE’s publication criteria as it currently stands. Therefore, we invite you to submit a revised version of the manuscript that addresses the points raised during the review process.

We look forward to receiving your revised manuscript.

Kind regards,

Hamid Sharifi

Academic Editor

PLOS ONE

Journal Requirements:

Reviewers' comments:

Reviewer's Responses to Questions

**Comments to the Author**

1. Is the manuscript technically sound, and do the data support the conclusions?

Reviewer #1: Yes

Reviewer #2: Partly

2. Has the statistical analysis been performed appropriately and rigorously? 

Reviewer #1: Yes

Reviewer #2: Yes

3. Have the authors made all data underlying the findings in their manuscript fully available?

Reviewer #1: Yes

Reviewer #2: Yes

4. Is the manuscript presented in an intelligible fashion and written in standard English?

Reviewer #1: Yes

Reviewer #2: Yes

5. Review Comments to the Author

Reviewer #1: Dear author,

I found your study topic to be of significance and the findings are of interest to the wider readership. The description of methods and discussion of the results are in adequate detail. I do have the following question and recommendation:

1) Is there a reason why work that does not involve sex has been described as "Straight work"? Is this an objective term? If not, I recommend using a more objective term/phrase.

2) Paragraph 3 under Conclusions (page 359) makes an inference about sex workers studied in this work not being a captive workforce. Did the present paper study whether the workers were captive or coerced? One of the objective ways to measure this would be interviewing using direct outreach, which was not carried out in this study. It also seems unlikely that the workers who are captive or are coerced, would be advertised online. So if this was not measured, I would advise against making inferences on this subject.

All the best.

Reviewer #2: I would like to express my appreciation for the effort devoted to the study on sex worker/sex worker time. This issue is extremely important and its approach provides a valuable contribution to the understanding of the labour rights of this population.The study on sex work time of sex workers is a significant contribution to academic literature, addressing a crucial issue that has historically been marginalized. However, a closer review highlights specific areas where methodology and the presentation of results can be strengthened to ensure the robustness and applicability of the conclusions.

Regarding the need for greater methodological clarity, it is suggested that the study could benefit from specifying the type of research design used. A qualitative, quantitative or mixed approach can influence the interpretation of the results and provide a deeper context about the nature of the phenomenon studied. Also, explicitly detailing the inclusion and exclusion criteria of the sample would allow readers to evaluate the representativeness of the studied population. Transparency in these areas is essential for the internal validity of the study.

As for the need to consider contextual variables, the research could benefit from a more holistic approach, integrating sociological theories that address the intersectionality of factors such as migration, criminalization, and unemployment in the lives of sex workers. The lack of specific classification of the sites and population studied may limit the generalization of the results. In this sense, a more detailed taxonomy could be extracted from methodologies used in previous studies that have addressed diversity in the sex work industry). In addition, the presentation of the self in everyday life could offer valuable perspectives for understanding social and work dynamics in specific environments.

In ethical terms, in the review of the report, questions arise related to the ethical issue, specifically regarding the management and detection of anomalous behaviors, such as trafficking in white women or child exploitation, in the framework of the research. I would like to have more details about the ethical protocols established to deal with these sensitive situations. In particular, I would like to know whether complaints were made to the competent authorities in cases where possible indicators of trafficking in women or child exploitation were detected. Understanding how these ethical issues were handled is essential to ensuring the integrity and social responsibility of the study.

In conclusion, the integration of these suggestions together with a more robust theoretical and methodological basis will strengthen the quality and relevance of the study, allowing a deeper and more nuanced understanding of sex workers' sex work time.

6. PLOS authors have the option to publish the peer review history of their article (what does this mean?). If published, this will include your full peer review and any attached files.

Reviewer #1: **Yes: **Nitya Kumar

Reviewer #2: No

---

## [Author Response · Author response to Decision Letter 0]

10 Jan 2024

PLOS ONE

Dear Dr. Kennedy,

Thank you for submitting your manuscript to PLOS ONE. After careful consideration, we feel that it has merit but does not fully meet PLOS ONE’s publication criteria as it currently stands. Therefore, we invite you to submit a revised version of the manuscript that addresses the points raised during the review process.

We look forward to receiving your revised manuscript.

Kind regards,

Hamid Sharifi

Academic Editor

PLOS ONE

Journal Requirements:

Reviewers' comments:

Reviewer's Responses to Questions

Comments to the Author

1. Is the manuscript technically sound, and does the data support the conclusions?

Reviewer #1: Yes

Reviewer #2: Partly

2. Has the statistical analysis been performed appropriately and rigorously?

Reviewer #1: Yes

Reviewer #2: Yes

3. Have the authors made all data underlying the findings in their manuscript fully available?

Reviewer #1: Yes

Reviewer #2: Yes

4. Is the manuscript presented in an intelligible fashion and written in standard English?

PLOS ONE does not copy edit accepted manuscripts, so the language in submitted articles must be clear, correct, and unambiguous. Any typographical or grammatical errors should be corrected at revision, so please note any specific errors here.

Reviewer #1: Yes

Reviewer #2: Yes

5. Review Comments to the Author

Reviewer #1: 

Dear author,

I found your study topic to be of significance and the findings are of interest to the wider readership. The description of methods and discussion of the results are in adequate detail. I do have the following question and recommendation:

1) Is there a reason why work that does not involve sex has been described as "Straight work"? Is this an objective term? If not, I recommend using a more objective term/phrase. - This was how respondents described their non-sex work employment in other studies. I have added clarification and mostly removed these.

2) Paragraph 3 under Conclusions (page 359) makes an inference about sex workers studied in this work not being a captive workforce. Did the present paper study whether the workers were captive or coerced? One of the objective ways to measure this would be interviewing using direct outreach, which was not carried out in this study. It also seems unlikely that the workers who are captive or are coerced, would be advertised online. So if this was not measured, I would advise against making inferences on this subject. - I have removed this sentence.

All the best. - Thank you for taking the time to review this work.

Reviewer #2: 

I would like to express my appreciation for the effort devoted to the study on sex worker/sex worker time. This issue is extremely important and its approach provides a valuable contribution to the understanding of the labour rights of this population.The study on sex work time of sex workers is a significant contribution to academic literature, addressing a crucial issue that has historically been marginalized. However, a closer review highlights specific areas where methodology and the presentation of results can be strengthened to ensure the robustness and applicability of the conclusions. - Thank you for taking the time to review this work.

Regarding the need for greater methodological clarity, it is suggested that the study could benefit from specifying the type of research design used. A qualitative, quantitative or mixed approach can influence the interpretation of the results and provide a deeper context about the nature of the phenomenon studied. Also, explicitly detailing the inclusion and exclusion criteria of the sample would allow readers to evaluate the representativeness of the studied population. Transparency in these areas is essential for the internal validity of the study. - Admittedly the sampling process is complex and I have made this more explicit by adding a flow diagram of the main steps in the Methods. The inclusion criteria are described as well: only advertisers who used at least one URL as a contact who were active in Aug/Sept 2022 were included. Tracked advertisers were those who represented collectives and provided information on individual workers. Workers counted were all those listed by advertisers on the advertisers’ sites.

As for the need to consider contextual variables, the research could benefit from a more holistic approach, integrating sociological theories that address the intersectionality of factors such as migration, criminalization, and unemployment in the lives of sex workers. The lack of specific classification of the sites and population studied may limit the generalization of the results. In this sense, a more detailed taxonomy could be extracted from methodologies used in previous studies that have addressed diversity in the sex work industry). In addition, the presentation of the self in everyday life could offer valuable perspectives for understanding social and work dynamics in specific environments. - Contextual variables are difficult to measure from archival materials therefore I have limited the discussion to what can be measured and avoid too much speculation about the data. Certainly there is a need for follow up work as I believe many of the shorter-term workers represented in the data have not been well studied. Also, I do classify the sites in the Working Collectively subsection of Results based on previous research. Most of these collectives are licensed businesses (83%) based on their website data. I mention another study based on this data that provides a more holistic view of the data in the Materials and Methods section (Kennedy, 2023). Interested readers are invited to read that for more information.

In ethical terms, in the review of the report, questions arise related to the ethical issue, specifically regarding the management and detection of anomalous behaviors, such as trafficking in white women or child exploitation, in the framework of the research. I would like to have more details about the ethical protocols established to deal with these sensitive situations. In particular, I would like to know whether complaints were made to the competent authorities in cases where possible indicators of trafficking in women or child exploitation were detected. Understanding how these ethical issues were handled is essential to ensuring the integrity and social responsibility of the study. - Keep in mind that this research is done under the guidance of workers who are currently active in the industry. The site used as the original ad source has strict policies against human trafficking and underage advertisers. Hence, none of the sampled advertisers used being under age as a marketing tool. While it is possible that there may have been workers who were coerced - which can be the case in other forms of work - there was no indication of this in the data. Of course, if there were unequivocal indicators of coercion I would have contacted the relevant authorities. Keep in mind there has been little if any work done to empirically validate whether trafficking can be identified using online materials. See for example (Boecking et al., 2015).

In conclusion, the integration of these suggestions together with a more robust theoretical and methodological basis will strengthen the quality and relevance of the study, allowing a deeper and more nuanced understanding of sex workers' sex work time. 

References

Boecking, B., Miller, K., Barnes, M., Boecking, B., & Kennedy, E. (2015). Leveraging publicly available data to discern patterns of human-trafficking activity. Journal of Human Trafficking, 1(1), 65–85. https://doi.org/10.1080/23322705.2015.1015342

Kennedy, L. (2023). Power users: Canadian sex workers’ use of technology post COVID. SocArXiv. https://doi.org/10.31235/osf.io/u5kd2

6. PLOS authors have the option to publish the peer review history of their article (what does this mean?). If published, this will include your full peer review and any attached files.

Do you want your identity to be public for this peer review? For information about this choice, including consent withdrawal, please see our Privacy Policy.

Reviewer #1: Yes: Nitya Kumar

Reviewer #2: No

---

## [Editor Report · Decision Letter 1]

26 Jan 2024

Estimating turnover and industry longevity of Canadian sex workers

PONE-D-23-35872R1

Dear Dr. Kennedy,

We’re pleased to inform you that your manuscript has been judged scientifically suitable for publication and will be formally accepted for publication once it meets all outstanding technical requirements.

Kind regards,

Hamid Sharifi

Academic Editor

PLOS ONE
---

## [Editor Report · Acceptance letter]

18 Mar 2024

PONE-D-23-35872R1 

PLOS ONE

Dear Dr. Kennedy, 

I'm pleased to inform you that your manuscript has been deemed suitable for publication in PLOS ONE. Congratulations! Your manuscript is now being handed over to our production team.

Kind regards, 

on behalf of

Dr. Hamid Sharifi 

Academic Editor

PLOS ONE